# Design and Synthesis of Polysiloxane Based Side Chain Liquid Crystal Polymer for Improving the Processability and Toughness of Magnesium Hydrate/Linear Low-Density Polyethylene Composites

**DOI:** 10.3390/polym12040911

**Published:** 2020-04-14

**Authors:** Xiaoxiao Guan, Bo Cao, Jianan Cai, Zhenxing Ye, Xiang Lu, Haohao Huang, Shumei Liu, Jianqing Zhao

**Affiliations:** 1School of Materials Science and Engineering, South China University of Technology, Guangzhou 510640, China; guanxiaoxiao.ok@163.com (X.G.); shawncb@126.com (B.C.); mscaijn@mail.scut.edu.cn (J.C.); mrzhxye@163.com (Z.Y.); hhhuang@scut.edu.cn (H.H.); liusm@scut.edu.cn (S.L.); 2Key Laboratory of Polymer Processing Engineering, Ministry of Education, Guangzhou 510640, China

**Keywords:** side chain liquid crystal polymer, magnesium hydroxide, low density polyethylene, toughness, processability

## Abstract

In this study, a polysiloxane grafted by thermotropic liquid crystal polymer (PSCTLCP) is designed and synthesized to effectively improve the processability and toughness of magnesium hydroxide (MH)/linear low-density polyethylene (LLDPE) composites. The obtained PSCTLCP is a nematic liquid crystal polymer; the liquid crystal phase exists in a temperature range of 170 to 275 °C, and its initial thermal decomposition temperature is as high as 279.6 °C, which matches the processing temperature of MH/LLDPE composites. With the increase of PSCTLCP loading, the balance melt torque of MH/LLDPE/PSCTLCP composites is gradually decreased by 42% at 5 wt % PSCTLCP loading. Moreover, the power law index of MH/LLDPE/PSCTLCP composite melt is smaller than 1, but gradually increased with PSCTLCP, the flowing activation energy of PSCTLCP-1.0 is lower than that of MH/LLDPE at the same shear rate, indicating that the sensitivity of apparent melt viscosity of the composites to shear rate and to temperature is decreased with the increase of PSCTLCP, and the processing window is broadened by the addition of PSCTLCP. Besides, the elongation at break of MH/LLDPE/PSCTLCP composites increases from 6.85% of the baseline MH/LLDPE to 17.66% at 3 wt % PSCTLCP loading. All the results indicate that PSCTLCP can significantly improve the processability and toughness of MH/LLDPE composites.

## 1. Introduction

Plastics modified by various inorganic fillers have been widely used, such as halogen-free flame-retarded polyolefins modified by Al (OH)_3_, Mg (OH)_2_ and others, polypropylene reinforced by talc, etc. [1,2,3]. However, when the inorganic filler loading reaches a certain level, the friction between the inorganic particles and the plastic matrix is exacerbated, the melt viscosity of thermoplastic composites is increased, and the processability is deteriorated [4,5,6]. Therefore, how to improve the processability of thermoplastic composites with high loading of inorganic fillers has attracted more and more attentions from academia, and especially from industry [7,8].

Adding processing aids is a usual method to improve the processability of thermoplastic composites [9,10]. As known, the linear polydimethylsiloxane (PDMS) is a good flexible polymer, and low molecular weight PDMS (silicone oil) is a good lubricant for plastics [11,12]. A small amount of polysiloxane with long chain alkyl side group obviously increased the melt flow rate and reduced the processing difficulty of thermoplastic composites [13]. Moreover, it was found that the toughness of plastics composites was enhanced with the increase of PDMS loading [14]. However, PDMS is a typical liquid polymer and cannot be conveniently used in industrial production, compared with powdered processing aids.

Thermotropic liquid crystal polymer (TLCP) possesses the excellent mechanical properties and lubrication for thermoplastic polymer melts [15,16]. During the melting process, the rigid TLCP macromolecules are easily oriented and arranged along the flow direction under the action of shear or tensile force, and ultimately improve the processability of thermoplastic composites [17,18,19,20].

However, for TLCP, the prerequisite for improving the processability of thermoplastic composites is that the liquid crystal phase transition temperature range is consistent with the processing temperature of thermoplastics [21]. Unfortunately, at present, the melting temperature of most main chain thermotropic liquid crystal polymers is quite high and is just comparable to that of the engineering plastics with high processing temperature [22], but is not well matched with the polyolefins with lower processing temperature [23]. Therefore, it has attracted more and more attention to reduce the melting point of the thermotropic liquid crystal polymer and further to expand its application in polyolefin-based thermoplastic composites with lower processing temperature [24]. Song et al. [25] prepared two series of wholly aromatic thermotropic copolysters containing 2-(a-phenylisopropyl) hydroquinone (PIHQ) moiety and found that the presence of bulky substituent of PIHQ unit reduced the melting temperature of aromatic thermotropic copolysters. Zhao et al. [26] prepared a new type of phosphorus-containing thermotropic liquid crystal copolyester with a flexible segment, and its liquid crystal phase existed in the range of 185 to 330 °C, which was matched with the processing temperature of most plastics. Xia et al. [23] synthesized a main chain liquid crystal copolyester with low melting point to match the processing temperature of polypropylene (PP), and it was found that the apparent melt viscosity of PP/TLCP composites was significantly reduced by the addition of the TLCP, compared with original PP. However, the main chain liquid crystal is mainly prepared by polycondensation reaction, and the preparation of large molecular weight polymers requires the high temperature, a complex purification process, and other special conditions. Accordingly, there are few main chain thermotropic liquid crystal polymers that are matched with polyolefins with the lower processing temperature. Moreover, the toughness of plastics composites is usually weakened by the addition of TLCP [27].

Polysiloxane grafted by thermotropic liquid crystal polymer (PSCTLCP) not only has the flexible PDMS segment as main chain, but also owns the liquid crystal performance of thermotropic liquid crystal polymer—although as the side chain [28,29,30]. The liquid crystal melting temperature of PSCTLCP was adjusted between 15 and 200 °C [31], and the isotropic temperature of PSCTLCP was changed from 84 [32] to 300 °C [33]. It indicates that the liquid crystal phase transition of PSCTLCP is well matched with the processing temperature of MH/LLDPE composites if a reasonably structured PSCTLCP is synthesized by the molecular design [34]. Besides, compared with PDMS, the powdery PSCTLCP is convenient to be used in industrial production. Therefore, PSCTLCP with reasonable structure it is likely to improve the processability and toughness of MH/LLDPE composites. However, to the authors’ knowledge, there has hardly been any report on simultaneously improving the processability and toughness of MH/LLDPE composites by PSCTLCP.

In this paper, a PSCTLCP with lower melting temperature matched with the processing temperature of MH/LLDPE/PSCTLCP composites is synthesized and characterized, and the effect of PSCTLCP loading on the processability and tensile properties of MH/LLDPE/PSCTLCP composites is studied. Besides, the modified mechanism of PSCTLCP on MH/LLDPE/PSCTLCP composites is investigated at different processing temperatures, rotation speeds, and shear rates to guide the actual processing in industry. Therefore, the work provides the feasible scheme to design and synthesize the well-structured liquid crystal polymer to meet the special needs in the plastic processing.

## 2. Experiment

### 2.1. Materials

Methylparaben, allyl chloride, 4-ethylbenzoic acid, and hydroquinone were purchased from Sain Chemical Technology (Shanghai, China) Co., Ltd. Thionyl chloride, 4-dimethylaminopyridine (DMAP), triethylamine ((Et)_3_N), and poly(methylhydrogeno)siloxane (PMHS, number-average molecular weight = 1900) were purchased from Aladdin Chemical Reagent Co., Ltd (Shanghai, China). Linear low-density polyethylene (LLDPE, DFDA 7042) was produced by Sinopec Guangzhou Co., Ltd (Guangzhou, China). Magnesium hydroxide (MH, HFR-30A) was produced by Shenzhen Hualin Chemical Co., Ltd. Pyridine, toluene, ethanol, acetone, N, N-dimethylformamide (DMF), tetrahydrofuran (THF) and methanol were purchased from Shenyang Chemical Co., Ltd (China).

### 2.2. Synthesis of 4-allyloxybenzoic Acid (AOBA)

The mixture of methylparaben (45.60 g, 0.3 mol), allyl chloride (30 mL, 0.37 mol), acetone (120 mL), and ethanol (30 mL) was heated to reflux temperature in a three-necked flask equipped with a magnetic stir bar and a condenser. Then, sodium hydroxide solution (50 wt %, 24 g) was slowly added into the above flask and the reaction lasted for 24 h at 50 °C. The reaction mixture was filtered by the vacuum suction and the solvent was removed by rotary evaporation. The residue is poured into sodium hydroxide solution (10 wt %, 180 mL) under stirring. After obtainment of a clear solution, dilute hydrochloric acid solution was slowly added until pH value of 4. The crude product was obtained by filtration, then washed with water several times, dried in a vacuum oven at 50 °C, further recrystallized from ethanol and vacuum dried again at 50 °C to obtain 4-allyloxybenzoic acid (AOBA) as white crystals. The synthesis route of AOBA is shown in Scheme 1.

### 2.3. Synthesis of 4-allyloxybenzoyl Chloride

AOBA (4.59 g, 0.02 mol), thionyl chloride (20 mL, 0.27 mol), and 4 drops of DMF were added in a single-mouth flask with an absorption instrument of hydrogen chloride. The reaction was stirred for 2 h at room temperature, then for 4 h at 80 °C to ensure the complete reaction. Afterwards, excess SOCl_2_ was distilled off under reduced pressure to obtain 4-allyloxybenzoyl chloride as pale yellow transparent liquid.

### 2.4. Synthesis of 4-hydroxyphenyl-4-ethylbenzoate (HPEB)

4-ethylbenzoic acid (50.0 g, 0.33 mol), thionyl chloride (50 mL, 0.68 mol), 4 drops of DMF were added into a one-necked flask with an absorption instrument of hydrogen chloride. The mixture was stirred for 5 h at room temperature, then the reaction was heated to reflux temperature and lasted for 2 h. The mixture was distilled under reduced pressure to obtain ethyl benzoyl chloride. Afterwards, hydroquinone (183.0 g, 1.66 mol) and pyridine (25 mL) were dissolved in THF (220 mL) to form a solution. The obtained ethyl benzoyl chloride was slowly added into the above solution in an ice bath under N_2_ atmosphere. The reaction was carried out for 2 h in ice bath, then was heated to reflux temperature and lasted for 8 h. Some solvents were distilled under reduced pressure, and the concentrate was poured into a beaker charged of a large amount of water, precipitated, filtered, washed with hot water several times, recrystallized from ethanol and vacuum dried overnight at 50 °C to obtain 4-hydroxyphenyl-4′-ethylbenzoate (HPEB) as white powder, and the synthetic process as shown in Scheme 1.

### 2.5. Synthesis of 4-ethylbenzoic Acid-4-allyloxybenzoic Acid Hydroquinone Diester (M)

HPEB (5.20 g, 0.02 mol), triethylamine (8 mL, 0.06 mol), DMAP (0.13g, 0.11 mmol) and dry THF (50 mL) were placed into a 100 mL three-necked flask with a magnetic stir bar under N_2_ atmosphere. The flask was equipped with a condenser in an ice bath. The obtained 4-allyloxybenzoyl chloride was slowly added into the above flask inside 30 min. Then, the reaction lasted for 24 h at room temperature. The reaction mixture was filtered, and the filtrate was poured into the acidified aqueous solution to obtain the precipitate. The precipitate was filtered and washed with several times with 5% potassium carbonate solution, deionized water and ethanol, respectively. The obtained solid was dried under vacuum at 50 °C for 24 h to obtain 4-ethylbenzoic acid-4′-allyloxybenzoic acid hydroquinone diester (M) as a white solid and the synthesis route is shown in Scheme 1.

### 2.6. Synthesis of PSCTLCP

M (5.04 g, 12.54 mmol) and PMHS (0.75 g, 0.04 mmol) were dissolved in toluene (50 mL), then 2.5 mL of H_2_PtCl_6_·6H_2_O/isopropyl alcohol (0.5 g of hexachloroplatinic acid hydrate dissolved in 100 ml of isopropyl alcohol) was slowly added into the above solution. The reaction lasted for 72 h at 65 °C under nitrogen and was monitored by FT-IR until the disappearance of the sharp vibrational band at 2165 cm^−1^ assigned as the Si-H stretching. The coarse product was obtained by precipitation of the reaction mixture from methanol, and further purified by dissolving in chloroform, precipitating from methanol several times in order to remove the excessive unreacted monomers. Finally, white powdery polymer (PSCTLCP) was obtained and the synthesis route as shown in Scheme 1.

### 2.7. Preparation of MH/LLDPE/PSCTLCP Composites

MH and LLDPE were dried in vacuum oven at 80 °C for 12 h prior to processing. The pre-mixed blend of LLDPE, MH and PSCTLCP was conducted using a mixer (HAAKE, RS600, Germany) at a temperature of 200 °C and screw speed of 50 rpm for 5 min. After being pulverized and dried, the well-mixed composite was injection-molded into tensile testing bars using a mini-injection system (Thermo Scientific, USA) at melt temperature of 210 °C and mold temperature of 30 °C. The compositions and code names of the samples were shown in Table 1.

### 2.8. Characterization

#### 2.8.1. Structural Characterization

The structure of AOBA, HPEB, M, PSCTLCP was characterized by Fourier transform infrared spectra (FT-IR) and ^1^H-NMR spectra. The FT-IR measurements were performed from 400 to 4000 cm^−1^ using a VERTEX70 spectrometer (Bruker, Karlsruhe, Germany) with KBr pellets. ^1^H-NMR measurements were recorded on a DMX-400 spectrometer (Bruker, Germany) with CDCl_3_ as the solvent and tetramethylsilane (TMS) as an internal standard.

#### 2.8.2. Differential Scanning Calorimetry (DSC)

Differential scanning calorimetry (DSC) measurements were performed using a Pekin-Elmer Diamond DSC (USA) at 10 °C /min from 30 to 300 °C in a nitrogen atmosphere. About 5 mg of each sample was sealed in an aluminum pan.

#### 2.8.3. Polarized Light Optical Microscopy (POM)

The optical texture of PSCTLCP was examined by a polarized light optical microscopy (POM) (Orthoglan, Leitz, Germany) equipped with a (STC200C, INTEC, USA) hot-stage.

#### 2.8.4. Thermogravimetric (TG)

Thermogravimetric analysis (TGA) was performed using a TG 209 F1 instrument (NETZSCH, Germany) at 10 °C /min from 40 to 800 °C in a nitrogen atmosphere. The samples (10 ± 1 mg) were placed in open platinum pans.

#### 2.8.5. Torque Rheometer

The balance torque of MH/LLDPE/PSCTLCP composite melt was determined by Haake torque XSS-300 rheometer (Thermo, Germany) at a rotation speed from 10 to 50 rpm. The test temperature was set at 170, 180, 195, 210, and 220 °C, respectively.

#### 2.8.6. High Pressure Capillary Rheometer

The rheological properties were measured by a high-pressure capillary rheometer (Rheologic 5000, Ceast, Italy) with a length-to-diameter ratio of 30/1. The tests were performed in a shear rate range from 10 to 5000 s^−1^ at 185, 195, 205, 215, and 225 °C, respectively.

#### 2.8.7. Scanning Electron Microscope (SEM)

Morphology and structure of MH/LLDPE/PSCTLCP composites were observed by scanning electron microscope (SEM, JEOL JSM-5900LV, Japan) equipped with an energy dispersive X-ray spectrometer (EDS, Oxford Isis, UK). The samples were fractured in liquid nitrogen, and the fracture surfaces were coated with gold to prevent charging on the surface.

#### 2.8.8. Tensile Testing

Tensile properties of MH/LLDPE/PSCTLCP composites were measured by an Instron 5967 model materials testing system (USA) according to ASTM D-638 standard. Samples of tension test were dumbbell-shaped and the direction of the tensile force was parallel to the length of samples.

## 3. Results and Discussion

### 3.1. Characterization of PSCTLCP

Figure 1a shows the FT-IR spectra of AOBA, HPEB, M, PSCTLCP. Compared with those of AOBA and HPEB, the broad and strong peak of –COOH disappears at 2550 to 3300 cm^−1^, the strong peak of -OH also disappears at 3500 cm^−1^ in FT-IR spectrum of M, meanwhile there is still the stretching vibration peak of CH_2_=CH– at 1643 cm^−1^ and the absorption peak of ester group at 1745 cm^−1^. The results indicate that M is successfully synthesized. For FT-IR spectrum of PSCTLCP, the stretching vibration peak of Si-H disappears at 2174 cm^−1^, the attribute peak of PMHS, and the stretching vibration peak of CH_2_=CH– is not found at 1643 cm^−1^ existed in FTIR spectrum of M. Besides, the main characteristic peak of Si-O-Si appears at 1000 to 1120 cm^-1^ with strong and wide bands. All the results indicate that the hydrosilylation reaction has taken place between PMHS and M [35].

Figure 1b shows ^1^H NMR spectra of M and PSCTLCP. The representative signals of the vinyl group in M at 5.28 and 6.0 ppm are not found in ^1^H NMR spectrum of PSCTLCP, which further indicates that the excessive M is completely removed and PSCTLCP is successfully obtained [36].

DSC experiment is used to examine the phase transition of PSCTLCP. The heating curve of PSCTLCP and the related data are showed in Figure 2 and Table 2, respectively. It is seen that the glass transition temperature (*T*_g_) of PSCTLCP is 34.5 °C. Furthermore, two heat absorption peaks appear in Figure 2. The temperature corresponding to the top value of the larger endothermic peak is 185.2 °C, that is the melt temperature (*T*_m_) of PSCTLCP. Another one presents smaller and appears at 277.7 °C, related to the liquid crystal phase change from anisotropic to isotropic state, and that is the isotropic temperature (*T*_i_) of PSCTLCP [37]. Besides, the initial melting temperature (*T*_initial_) PSCTLCP is 173.6 °C.

The polarizing microscope (POM) with a hot stage is used to visually observe the liquid-crystalline transition and optical textures, and Figure 3 shows the POM images at different temperatures. It is found that PSCTLCP begins to transfer from the solid state to liquid state at 170 °C, very close to *T*_initial_ from DSC test, and the nematic texture of polymer is observed in Figure 3a. The higher temperature, the more obvious texture morphology and color are observed [38]. In Figure 3b, the texture of marbled nematic liquid crystal is seen in bright field at 210 °C. When the temperature is continuously raised to 275 °C, the field of vision is gradually darkened and the liquid crystal phase gradually disappears, indicating PSCTLCP transfers from the anisotropic liquid to isotropic transparent liquid and *T*_i_ is about 275 °C, in line with the result from DSC experiment.

Figure 4 shows the thermal decomposition process of PSCTLCP under nitrogen atmosphere. It is found that T_5_ (decomposition temperature at 5 wt % mass loss) and T_50_ (at 50 wt % mass loss) are 279.6 and 429.8 °C, respectively, indicating the quite good thermal stability of obtained PSCTLCP.

As we know, the processing temperature of MH/LLDPE composites is usually between 160~250 °C, which is well matched with the melt temperature (*T*_m_ of 185.2 °C), the isotropic temperature (*T*_i_ of 277.7 °C) and thermal decomposition temperature (T_5_ of 279.6 °C). Therefore, the obtained PSCTLCP is suitably used as a processing aid to improve the processability of MH/LLDPE composites.

### 3.2. Processability of MH/LLDPE/PSCTLCP Composites

Torque rheometer is an important instrument to characterize the processability of polymer materials. Figure 5 shows the evolution curves of melt torque values of MH/LLDPE/PSCTLCP composites at 170 °C and the rotation speed of 50 rpm. The balance torque of the MH/LLDPE/PSCTLCP composites is gradually decreased with the increase of PSCTLCP content. Compared with that of MH/LLDPE composite without PSCTLCP (16.59 N·m), the balance torque of MH/LLDPE/PSCTLCP composites is decreased by 15% (14.13 N·m), 18% (13.69 N·m), 29% (11.86 N·m), and 42% (9.67 N·m), respectively, for the samples containing PSCTLCP of 0.6, 1, 3, 5 wt %, respectively. The results indicate that PSCTLCP significantly reduces the melt viscosity and enhances the processability of the composites. After melting, PSCTLCP enters a liquid-like “liquid crystal state” and is easily oriented into microfibers in the shear direction during processing [39], and is regarded as a lubricant, which can reduce the entanglement between LLDPE molecular chains and the aggregation between MH particles, thus reducing the melt viscosity and improving the processability of MH/LLDPE/PSCTLCP composites [40].

In order to investigate the effect of the external conditions, such as stress and temperature, on the usability of PSCTLCP, samples PSCTLCP-1.0 and MH/LLDPE are selected as a partner to be compared the melt torque at different temperatures and rotation speeds. The relationship between balance torque of composites and rotation speed at 195 °C is showed in Figure 6a. It is seen that the balance torque of PSCTLCP-1.0 is higher than that of MH/LLDPE at lower rotation speed, while the former is significantly smaller at higher rotation speed, and their torque difference gradually becomes bigger as the increasement of rotation speed. It is deduced that the molecules of PSCTLCP are oriented in the shear direction during the melt processing when the enough shear force exists. The larger the shear force, the easier the orientation of PSCTLCP and the better the processability [21,22,23]. Figure 6b shows the balance torque vs. temperature curves for MH/LLDPE and PSCTLCP-1.0 at 50 rpm. With the increasing temperature, the balance torque of MH/LLDPE and PSCTLCP-1.0 is significantly decreased. And the balance torque of PSCTLCP-0.1 is significantly lower than that of MH/LLDPE at the same temperature, indicating that the processability of PSCTLCP-1.0 is superior to that of MH/LLDPE in the whole processing temperature range. When the temperature rises from 170 to 220 °C, the balance torque of MH/LLDPE and PSCTLCP-1.0 is decreased by 32.7% and 18.7%, respectively. It indicates that the processing sensitivity of MH/LLDPE composites to the temperature is reduced by the addition of PSCTLCP.

The above results show that the processability of MH/LLDPE/PSCTLCP composites is not only affected by PSCTLCP content, but also by the rotation speed (shear force) and processing temperature, which is attributed to the orientation of PSCTLCP during the processing [41].

In addition, the capillary rheometer is also used to characterize the processability of MH/LLDPE/PSCTLCP composites. Figure 7 shows the relationship between apparent viscosity of the MH/LLDPE/PSCTLCP composites and shear rate, and between shear stress and shear rate at 195 °C. It is seen that the apparent viscosity and shear stress of composites are all gradually decreased with the increase of PSCTLCP content in the experimental range of shear rate, which further proves that the processability of MH/LLDPE/PSCTLCP composites is significantly improved by the addition of PSCTLCP [42].

Sample PSCTLCP-1.0 is selected to further exhibit the processing temperature on the rheological property. The logarithmic plots of shear stress (τ) versus shear rate (γ) for PSCTLCP-1.0 at 185, 205 and 225 °C are shown in Figure 8a. The logarithm of shear stress is linearly related to that of shear rate, suggesting that the melt flow of PSCTLCP-1.0 follows the power-law equation under experimental conditions [42]. According to the power-law equation (τ = *K*·γ*^n^*), the power law index (*n*) of MH/LLDPE/PSCTLCP composites with different PSCTLCP loadings is calculated. As shown in Figure 8b, all the *n* values of MH/LLDPE/PSCTLCP composites are less than 1, it means that the melt of MH/LLDPE/PSCTLCP composites behaves a pseudoplastic fluid. Furthermore, the *n* value of MH/LLDPE/PSCTLCP composites is increased from 0.38 to 0.53 when the content of PSCTLCP is increased from 0 to 5 wt %. It indicates that the non-Newtonian property of MH/LLDPE/PSCTLCP composites is weakened with the increasing of PSCTLCP loading. The results show that the sensitivity of apparent melt viscosity for composites to shear rate is decreased and the processing window is broadened with the increase of PSCTLCP.

According to the Arrhenius equation (η*_a_* = Aexp(*E*/RT)), the activation energy (*E*) of the composite melt is calculated by means of linear regression [6,7,43,44]. Figure 9a,b show the relationship between the logarithm of apparent viscosity (ln*η_a_*) and the reciprocal of absolute temperature (1/T) for MH/LLDPE and PSCTLCP-1.0, respectively. ln*η_a_* is increased with 1/T, indicating that the viscosity of the composite melt is gradually decreased with the increase of the processing temperature. With the increase of temperature, the mobility of the molecular chains is increased, and the interaction force between the molecular chains is reduced, thereby the viscosity of the composite is reduced and the fluidity of the composite melt is improved [43]. Furthermore, the values of *E* and the linear correlation coefficient (*R*_0_) are obtained in Table 3. It is found that *E* values of both MH/LLDPE and PSCTLCP-1.0 are continuously decreased with the increase of shear rate, which is attributed to the disentanglement between molecular chains at high shear rate [44,45]. On the other hand, the *E* value of PSCTLCP-1.0 is lower than that of MH/LLDPE at the same shear rate, indicating that PSCTLCP-1.0 melt is easier to flow compared to MH/LLDPE melt, and only 1 wt % PSCTLCP is obvious to improving the processability of MH/LLDPE composites.

### 3.3. The Facture Morphology of MH/LLDPE/PSCTLCP Composites

SEM test is used to study the effect of PSCTLCP on the facture morphology of MH/LLDPE/PSCTLCP composites, and the results are shown in Figure 10. According to the basic structure of composite materials [4,19], LLDPE is the continuous phase, MH is the dispersed phase and PSCTLCP should be located at the surface between MH and LLDPE. Some labels are used to clearly show MH and PSCTLCP in Figure 10. Moreover, the mapping scattering images of MH particles and PSCTLCP fibers in PSCTLCP-1.0 are shown in Figure 10d–f. Due to the poor compatibility between PSCTLCP and MH/LLDPE composites, liquid crystal fibers are pulled out from LLDPE matrix and thereby some holes are formed in fracture surface, as shown in Figure 10b,c. It is seen that the number of holes is increased with increasing PSCTLCP loading. The more liquid crystal fibers are formed, the better processability of composites [18,19,20]. The results further explain that the processability of MH/LLDPE/PSCTLCP composites is significantly improved by the addition of PSCTLCP.

### 3.4. Mechanical Properties of MH/LLDPE/PSCTLCP Composites

The tensile properties of MH/LLDPE/PSCTLCP composites are shown in Figure 11. Due to the flexibility of the main chains of PSCTLCP molecules, the tensile strength and tensile modulus of composites are slightly decreased with PSCTLCP loading. Meanwhile, the elongation at break of composites is significantly increased, as shown in Figure 11c. Compared with 6.85% of MH/LLDPE composite without PSCTLCP, the elongation at break of MH/LLDPE/PSCTLCP composites is increased to 10.68%, 15.8%, and 17.66%, respectively, and the corresponding loading of PSCTLCP is 0.6, 1, 3 wt %, respectively. The results mean the toughness of MH/LLDPE/PSCTLCP composites is enhanced by the addition of PSCTLCP, most likely due to the polysiloxane structure in the macromolecular chains of PSCTLCP, similar to the toughness enhancement of composites by the addition of polydimethylsiloxane [14].

## 4. Conclusions

In order to match the processing temperature and improve the processability of MH/LLDPE composites, polysiloxane grafted by thermotropic liquid crystal polymer (PSCTLCP) was successfully synthesized. The balance melt torque of MH/LLDPE/PSCTLCP composites was obviously decreased with the loading of PSCTLCP and that of the composite (5.0 wt % loading) was decreased from 16.59 N·m of the baseline MH/LLDPE to 9.67 N·m. Besides, PSCTLCP weakened the non-Newtonian property and decreased the flowing activation energy of MH/LLDPE/PSCTLCP composites, and thus broadened the processing window and improved the processability. Considering the flexible polysiloxane as main chains of PSCTLCP, the elongation at break of MH/LLDPE composites was significantly increased, and the tensile strength and modulus were slightly decreased with PSCTLCP loading.

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
