# Peer review of "Design and Synthesis of Polysiloxane Based Side Chain Liquid Crystal Polymer for Improving the Processability and Toughness of Magnesium Hydrate/Linear Low-Density Polyethylene Composites"

_polymers, 2020, doi:10.3390/polym12040911_

Round 1

Reviewer 1 Report

Guan et al. in this work synthesized a side chain lyotropic liquid crystal polysiloxane grafted with mesogens (PSCTLCP) which was used as an additive to enhance the processability and toughness of magnesium hydrate/linear low density polyethylene composites. The results show that the effects of PSCTLCP are rather positive to the processing of the composites though the tensile strength and tensile modulus of the composites are slightly decreased with PSCTLCP loading. The work can make contributions to the manufacturing of low processing temperature polymer/inorganic composites. The manuscript can be published after the following comments are addressed.

  1. Figure 2, the determination of Tg is questionable. If the DSC test was started at 25 C, there would be an unstable region around 25 C. In other words, 34.5 C is not the Tg. I suggest the DSC data should cover the range between -60 to 250 C to determine Tg and other transitions.
  2. The endothermic peak at 185.2 C should be the melting point. However, the exothermic peak at 277.7 C is definitely not the Ti because the isotropization transition should be endothermic, not exothermic. In Figure 3, the decomposition temperature at 5 wt% mass loss (T5) is 279.6 C. It is apparent that the exothermic peak at 277.7 C is caused by degradation, not isotropization. The interpretation of the DSC data should be re-checked. The Ti may be around 220 C where a small endothermic hump can be seen.
  3. Figure 10, it is difficult to distinguish the MH, LLDPE, and PSCTLCP in the three SEM images. The authors may use labels to clearly show the compositions in the images.
  4. Figure 11, the data of PSCTLCP-5.0 are not shown. It is expected that the tensile strength and tensile modulus of PSCTLCP-5.0 will be lower than those of others. How about the elongation at break? Keep increasing or deteriorate for PSCTLCP-5.0?

Author Response

Point 1: Figure 2, the determination of Tg is questionable. If the DSC test was started at 25 oC, there would be an unstable region around 25 oC. In other words, 34.5 oC is not the Tg. I suggest the DSC data should cover the range between -60 to 250 oC to determine Tg and other transitions.

Response 1: We appreciate Reviewer’s valuable comment. We have redrawn the DSC curve of PSCTLCP, shown in Figure 2. There is a turning point of endothermic process with the increase of temperature from 30 oC to 45 oC for PSCTLCP , and the heat absorption of PSCTLCP gradually decreases after the turning point, which indicates that the polymer chain segments begin to move, that is the polymer chain enters into the  high-elastic state from the glassy state and the temperature corresponding to the turning point is the glass transition temperature of the polymer. Besides, the obtained PSCTLCP is white glassy powder at room temperature (near 25 oC), that is the Tg of PSCTLCP must not be below 25 oC. In fact, the shape of DSC curve is very similar with the one of LCP reported in the references as followings. Bubnov, A.; Vladimira, N.; Pociecha, D. A Liquid-Crystalline Co-Polysiloxane with Asymmetric Bent Side Chains. Macromol. Chem. Phys. 2011, 212(2), 191-197. Yao, W.; Gao, Y.; Zhang C. A series of novel side chain liquid crystalline polysiloxanes containing cyano- and cholesterol-terminated substituents: Where will the structure-dependence of terminal behavior of the side chain reappear?. J. Polym. Sci. Pol. Chem. 2017, 55(10), 1765-1772.

Point 2: The endothermic peak at 185.2 oC should be the melting point. However, the exothermic peak at 277.7 oC is definitely not the Ti because the isotropization transition should be endothermic, not exothermic. In Figure 3, the decomposition temperature at 5 wt% mass loss (T5) is 279.6 oC. It is apparent that the exothermic peak at 277.7 oC is caused by degradation, not isotropization. The interpretation of the DSC data should be re-checked. The Ti may be around 220 oC where a small endothermic hump can be seen.

Response 2: We appreciate Reviewer’s valuable comment. If the Ti of PSCTLCP is around the 220 oC, it cannot explain that Ti observed by POM is around 275 oC. It is found from Fig.2 that there exists a small endothermic hump around 277.7 oC, and it is reasonable to believe 277.7 oC as the Ti of PSCTLCP, which is very close to the Ti from the observation by POM. The small endothermic hump around 220 oC may be an instrumental error. In fact, the shape of DSC curve is very similar with the one of LCP reported in the references as followings. Yao, W.; Gao, Y.; Li, F. Influence of shorter backbone and cholesteric monomer percentage on the phase structures and thermal-optical properties of linear siloxane tetramers containing cholesterol and benzene methyl ether groups. Rsc Adv. 2016, 6(90), 87502-87512. Wang, J.W.; Meng, F.B.; Li, Y.H. Synthesis and characterization of side-chain cholesteric liquid-crystalline polysiloxanes containing different space groups. J. Appl. Polym. Sci., 2009, 111(4), 2078-2084.

Point 3: Figure 10, it is difficult to distinguish the MH, LLDPE, and PSCTLCP in the three SEM images. The authors may use labels to clearly show the compositions in the images.

Response 3: We appreciate Reviewer’s valuable comment. According to some references (refs.4,19), some labels have been used to clearly show the compositions in the SEM images. Moreover, SEM in this manuscript is mainly used to show how many holes are formed after the liquid crystal fibers are pulled out from LLDPE matrix. The more liquid crystal fibers are formed, the better processability of composites.

Point 4: Figure 11, the data of PSCTLCP-5.0 are not shown. It is expected that the tensile strength and tensile modulus of PSCTLCP-5.0 will be lower than those of others. How about the elongation at break? Keep increasing or deteriorate for PSCTLCP-5.0?

Response 4: We appreciate Reviewer’s valuable comment. Due to the flexible structure of polysiloxane in the macromolecular chains of PSCTLCP, the elongation at break of MH/LLDPE/PSCTLCP composites is enhanced by the addition of PSCTLCP. We think that the elongation at break of PSCTLCP-5.0 will larger than that of PSCTLCP-3.0.

Reviewer 2 Report

Title: Design and synthesis of polysiloxane based side chain liquid crystal polymer for improving the processability and toughness of magnesium hydrate/linear low density polyethylene composites

Authors: Xiaoxiao Guan and colleagues

Overall assessment:

The authors investigated various properties of the polymeric materials arbitrarily and described the testing results often tediously and casually. Such irrelevant descriptions prevent the readers from the appropriate appreciation of this study. Perhaps the tediousness and casualties are because of the lack of the objective and careful readings before the submission. To continue the reviewing process, the authors must elaborate the manuscript thoroughly.

Followings are the comments that will be helpful for the resubmission:

Specific comments:

  1. The abstract is extremely lengthy. The authors must compress the abstract into that with 200 words maximum.

  1. The English seems to be clumsy; therefore, I strongly recommend the authors to offer the edition of the English to some office.

  1. The introduction section is tedious to read because some unnecessary descriptions are contained. Additionally, although various abbreviations are used, they are not listed using a nomenclature section. Because of these irrelevancies, the introduction section requires a high concentration of the readers. The authors must describe the introduction more clearly while reducing the unnecessary information for the purpose of this study.

  1. Despite the tediousness of the introduction section, the experimental section is often too terse. It is more important for many readers to reproduce the experiment than to understand the motivation of the authors for conducting this study.

  1. The subsection 2.8 is extremely too terse. The authors provide the independent subsections for the syntheses of AOBA, HPEB, M, and PSCTLCP as 2.2-2.5, respectively. In contrast, the experimental characterizations are treated in one subsection. It seems to me that the authors regard his style of the descriptions

  1. The synthesis of 4-allyloxybenzoyl chloride was not conducted, was it?

  1. The definition of R0 is missing.

  1. There are no descriptions on the methods how to obtain the E and R values.

  1. It is difficult to image the composite materials fabricated in this study. I strongly recommend the authors to show several photographs of the composites.

  1. There is a possibility to produce the anisotropy by the extrusion and granulation. Was there an anisotropy in the fabricated composites? If the composite materials were anisotropic, the direction where the force was applied in the tension test must be denoted.

  1. Commonly to the issue in Comment 9, there are no descriptions and figures on the configuration of the composite materials used for the experimental characterizations.

  1. I cannot find any descriptions on the results of the experimental characterization on the toughness of the materials.

Recommendation

Thorough revisions are required and the authors must prepare the revised version as well as the point-by-point response to the comments described above. The tedious and casual descriptions often prevent the readers from the appreciation of the essence of this study; therefore, the authors must improve the manuscript by careful and objective readings. I’d like to receive a revised version in due course.

Author Response

Point 1: The abstract is extremely lengthy. The authors must compress the abstract into that with 200 words maximum.

Response 1: We appreciate Reviewer’s valuable comment. We have rewritten the abstract in the revised manuscript.

Point 2: The English seems to be clumsy; therefore, I strongly recommend the authors to offer the edition of the English to some office.

Response 2: We appreciate Reviewer’s valuable comment. We have checked the grammar, spelling, and etc., carefully and polished the writing by a native speaker.

Point 3: The introduction section is tedious to read because some unnecessary descriptions are contained. Additionally, although various abbreviations are used, they are not listed using a nomenclature section. Because of these irrelevancies, the introduction section requires a high concentration of the readers. The authors must describe the introduction more clearly while reducing the unnecessary information for the purpose of this study.

Response 3: We appreciate Reviewer’s valuable comment. We have deleted some unnecessary descriptions and reduced the number of abbreviations to make the introduction more clearly for readers. Moreover, we have added the Abbreviations behind the Conclusion.

Point 4: Despite the tediousness of the introduction section, the experimental section is often too terse. It is more important for many readers to reproduce the experiment than to understand the motivation of the authors for conducting this study.

Response 4: We appreciate Reviewer’s valuable comment. In order to reproduce the experiment for many readers, we have rewritten the experimental section to give more experiment information.

Point 5: The subsection 2.8 is extremely too terse. The authors provide the independent subsections for the syntheses of AOBA, HPEB, M, and PSCTLCP as 2.2-2.5, respectively. In contrast, the experimental characterizations are treated in one subsection. It seems to me that the authors regard his style of the descriptions

Response 5: We appreciate Reviewer’s valuable comment. We have checked the experimental section and especially rewritten the subsection 2.8 and the experimental characterizations to give more experiment information.

Point 6: The synthesis of 4-allyloxybenzoyl chloride was not conducted, was it?

Response 6: No, 4-allyloxybenzoyl chloride was synthesized in 2.3.

Point 7: The definition of R0 is missing.

Response 7: We appreciate Reviewer’s valuable comment. The R0 is the linear correlation coefficient. The definition of R0 is shown in line 322 with colored characters in the revised manuscript.

Point 8: There are no descriptions on the methods how to obtain the E and R values.

Response 8: We appreciate Reviewer’s valuable comment. According to the Arrhenius equation (ηa=Aexp(E/RT)), R is the universal gas constant,the values of constants E and R0 can be determined by means of linear regression (refs. 6-7,43-44) and the fitted curve of composites are shown in Figure 9.

Point 9: It is difficult to image the composite materials fabricated in this study. I strongly recommend the authors to show several photographs of the composites.

Response 9: We appreciate Reviewer’s valuable comment. The composite materials are fabricated by traditional melt blending. According to common sense of composite materials (refs.4,19), LLDPE is the continuous phase, MH is the dispersed phase and PSCTLCP should be located at the surface between MH and LLDPE. However, in this paper, we mainly focus on the processability and toughness of composites. In fact, SEM test is used to study the effect of PSCTLCP on the facture morphology of MH/LLDPE/PSCTLCP composites.

Point 10: There is a possibility to produce the anisotropy by the extrusion and granulation. Was there an anisotropy in the fabricated composites? If the composite materials were anisotropic, the direction where the force was applied in the tension test must be denoted.

Response 10: We appreciate Reviewer’s valuable comment. The melted PSCTLCP is anisotropic and is easily oriented under the external forces during processing. Samples to be tested are obtained by an injection molding machine at melt temperature of 210 oC. We have added the corresponding description about the direction where the force is applied in the tension test in the characterization section.

Point 11: Commonly to the issue in Comment 9, there are no descriptions and figures on the configuration of the composite materials used for the experimental characterizations.

Response 11: We appreciate Reviewer’s valuable comment. SEM test is used to study the effect of PSCTLCP on the facture morphology of MH/LLDPE/PSCTLCP composites.

Point 12: I cannot find any descriptions on the results of the experimental characterization on the toughness of the materials.

Response 12: The elongation at break of MH/LLDPE/PSCTLCP composites is used to characterize the toughness of the materials. The elongation at break of composites is significantly increased, shown in Figure 11(c).

Reviewer 3 Report

Guan et al. reported the manuscript entitled “Design and Synthesis of Polysiloxane based Side Chain Liquid Crystal Polymer for Improving the Processability and Toughness of Magnesium Hydrate/Linear Low Density Polyethylene Composites’’ with detailed FTIR, DSC, TGA, NMR, rheological and mechanical properties, and morphology. This manuscript needs major revision before publication. 

  1. Abstract should be more quantitative.
  2. In the introduction part, the novelty of this work should be highlighted in the last paragraph.
  3. The introduction section should be more informative with some blend-based composites articles. The author should cite the following papers (a) Graphene nanoplatelet-reinforced poly (vinylidene fluoride)/high-density polyethylene blend-based nanocomposites with enhanced thermal and electrical properties. (b) Enhanced thermal stability, toughness, and electrical conductivity of carbon nanotube-reinforced biodegradable poly(lactic acid)/poly(ethylene oxide) blend-based nanocomposites
  4. In line 122, please change from 30ml to 30 ml.
  5. In line 135, change from .27mol to .27 mol. In line 137, please change from 2 hours to 2 h. The authors need to check carefully the whole manuscript.
  6. In line 174, change from 80 oC to 80 °C. Maintain consistency throughout the manuscript.
  7. The author should explain why there is a broad peak at 225 °C? If possible please provide the DSC cooling curve.
  8. Regarding the TGA part, the author should give TGA degradation curves of the MH/LLDPE/PSCTLCP composites.
  9. In Fig. 10, please highlight the location of PSCTLCP by the arrow mark. Please provide EDS of the fabricated samples.

Author Response

Point 1: Abstract should be more quantitative.

Response 1: We appreciate Reviewer’s valuable comment. We have rewritten the abstract in the revised manuscript.

Point 2: In the introduction part, the novelty of this work should be highlighted in the last paragraph.

Response 2: We appreciate Reviewer’s valuable comment. We have rewritten the last paragraph to show the novelty of this work in the revised manuscript.

Point 3: The introduction section should be more informative with some blend-based composites articles. The author should cite the following papers (a) Graphene nanoplatelet-reinforced poly (vinylidene fluoride)/high-density polyethylene blend-based nanocomposites with enhanced thermal and electrical properties. (b) Enhanced thermal stability, toughness, and electrical conductivity of carbon nanotube-reinforced biodegradable poly(lactic acid)/poly(ethylene oxide) blend-based nanocomposites

Response 3: We appreciate Reviewer’s valuable comment. We have cited these articles in the revised version.

Point 4: In line 122, please change from 30ml to 30 ml.

Response 4: We appreciate Reviewer’s valuable comment. In line 104, 30ml has changed to 30 mL in the revised manuscript.

Point 5: In line 135, change from .27mol to .27 mol. In line 137, please change from 2 hours to 2 h. The authors need to check carefully the whole manuscript.

Response 5: We appreciate Reviewer’s valuable comment. These mistakes have been corrected in the revised manuscript. In line 117, .27mol has changed to .27 mol. In line 119, 2 hours has changed to 2 h. We also carefully examined and corrected some similar mistakes in the revised manuscript. For example, 4 hours has changed to 4 h in line 119. In line 123, .33mol has changed to .33 mol.

Point 6: In line 174, change from 80 oC to 80 oC. Maintain consistency throughout the manuscript.

Response 6: We appreciate Reviewer’s valuable comment. In line 155, 80 oC has changed to 80 oC. Many similar errors have been corrected in the revised manuscript. For example, in line 183, 195 oC has changed to 195 oC.

Point 7: The author should explain why there is a broad peak at 225 oC? If possible please provide the DSC cooling curve.

Response 7: We appreciate Reviewer’s valuable comment. DSC experiment is used to examine the phase transition of PSCTLCP. In fact, the phase transition temperatures of PSCTLCP from POM and DSC are very close to each other, indicating that the results about phase transition of PSCTLCP is credible. The broad peak at 225 oC may be an instrumental error. It would be better to provide the DSC cooling curve. Unfortunately, due to the virus of COVID-19, our laboratories are still closed. we cannot provide the DSC cooling curve in the revised manuscript within 10 days.

Point 8: Regarding the TGA part, the author should give TGA degradation curves of the MH/LLDPE/PSCTLCP composites.

Response 8: Thank you for your suggestion. As we all know, the T5 (decomposition temperature at 5 wt% mass loss) of LLDPE is over 300 oC, the T5 of MH as inorganic nanoparticles is higher than that of LLDPE. In our work, the TGA test results of PSCTLCP in Figure 4 indicate that the T5 of PSCTLCP is 279.6 oC, lower than those of LLDPE and MH. Therefore, the thermostability of MH/LLDPE/PSCTLCP composites depends mainly on PSCTLCP. In fact, the aim to test TGA of PSCTLCP lies in estimating whether PSCTLCP is thermostable during the processing of MH/LLDPE/PSCTLCP composites and the result shows that the T5 of PSCTLCP is higher than the processing temperature.

Point 9: In Fig. 10, please highlight the location of PSCTLCP by the arrow mark. Please provide EDS of the fabricated samples.

Response 9: We appreciate Reviewer’s valuable comment. According to some references (refs.4,19), we have highlighted the location of PSCTLCP by the arrow mark in Figure 10. Moreover, the EDS of PSCTLCP-1.0 have been added in Figure 10.

Round 2

Reviewer 1 Report

  1. In the DSC Figure, the steep change of heat between 30 oC to 45 oC is the initial unstable data, which occurs for every DSC heating scan. It is not possible to determine any transition in this range. The DSC data in Figure 2 don’t look like the ones in the references the authors provided in the response. It is easy to redo the DSC starting from a lower temperature. I don’t know why the authors are not willing to do so.
  2. The decomposition temperature at 5 wt% mass loss (T5) is 279.6 oC, around which a combination of multiple endothermic and exothermic reactions may occur due to the degradation of the polymer. It is not meaningful to determine the physical phase transition around this temperature. If the small endothermic hump around 220 oC is an instrumental error as claimed by the authors, redo the DSC to eliminate it. Also, the DSC data in Figure 2 don’t look like the ones in the references the authors provided in the response.
  3. In Figure 11, can the authors show the data of PSCTLCP-5.0? Not just speculate in the response.

Author Response

Point 1: In the DSC Figure, the steep change of heat between 30 oC to 45 oC is the initial unstable data, which occurs for every DSC heating scan. It is not possible to determine any transition in this range. The DSC data in Figure 2 don’t look like the ones in the references the authors provided in the response. It is easy to redo the DSC starting from a lower temperature. I don’t know why the authors are not willing to do so.

Response 1: We appreciate Reviewer’s valuable comment. It would be better to redo the DSC starting from a lower temperature. Unfortunately, due to the virus of COVID-19, our laboratories are still closed. Besides, the similar method of selecting Tg has been reported by Yao and Bubnov et al. and Tg of LCP was also determined by the initial unstable data. (Yao et al., A series of novel side chain liquid crystalline polysiloxanes containing cyano- and cholesterol-terminated substituents: Where will the structure-dependence of terminal behavior of the side chain reappear? J. Polym. Sci. Pol. Chem. 2017, 55(10), 1765-1772. Bubnov et al., A Liquid-Crystalline Co-Polysiloxane with Asymmetric Bent Side Chains. Macromol. Chem. Phys. 2011, 212(2), 191-197.).

Point 2: The decomposition temperature at 5 wt% mass loss (T5) is 279.6 oC, around which a combination of multiple endothermic and exothermic reactions may occur due to the degradation of the polymer. It is not meaningful to determine the physical phase transition around this temperature. If the small endothermic hump around 220 oC is an instrumental error as claimed by the authors, redo the DSC to eliminate it. Also, the DSC data in Figure 2 don’t look like the ones in the references the authors provided in the response.

Response 2: We appreciate Reviewer’s valuable comment. As we all know, Ti and T5 of some TLCP are very close to each other (Zhang et al., Synthesis and properties of side chain cholesteric liquid-crystalline polyacrylates containing two mesogenic groups. J. Appl. Polym. Sci. 2003, 88(8),1936-1941. Sainath et al., Synthesis, characterization and liquid crystalline properties of polyacrylates and polymethacrylates containing aryl ester pendant unit. J. Appl. Polym. Sci. 2000, 75(4), 465-474. Amer et al., Liquid-crystalline azobenzene-containing ferrocene-based polymers: Study on synthesis and properties of main-chain ferrocene-based polyesters with azobenzene in the side chain. Polym. Adv. Technol. 2013, 24(2), 181-190.). Moreover, the Ti is not only examined by DSC, but also determined by POM. It is reasonable to believe 277.7 oC as the Ti of PSCTLCP by DSC, which is very close to the Ti (275 oC) from the observation by POM. It would be better to redo the DSC. Unfortunately, due to the virus of COVID-19, our laboratories are still closed.

Point 3: In Figure 11, can the authors show the data of PSCTLCP-5.0? Not just speculate in the response.

Response 3: We appreciate Reviewer’s valuable comment. The elongation at break of PSCTLCP-5.0 is 19.43%, which is larger than that of PSCTLCP-3.0.

Reviewer 2 Report

The manuscript is adequately revised; therefore, I can recommend the revised version to be published as it is.

Author Response

Point 1: The manuscript is adequately revised; therefore, I can recommend the revised version to be published as it is.

Response 1: We appreciate Reviewer’s valuable comment. Thanks very much!
